# LOSS META-LEARNING FOR FORECASTING

## ABSTRACT

Meta-learning of loss functions for supervised learning has been used to date for classification tasks, or as a way to enable few-shot learning. In this paper, we show how a fairly simple loss meta-learning approach can substantially improve regression results. Specifically, we target forecasting of time series and explore case studies grounded on real-world data, and show that meta-learned losses can benefit the quality of the prediction both in cases that are apparently naive, and in practical scenarios where the performance metric is complex, time-correlated, non-differentiable, or not known a priori.

## 1 INTRODUCTION

Loss functions drive the training process of supervised machine learning models. In the vast majority of cases, loss functions are designed to be *generic* enough to work well with a wide range of application scenarios. In regression problems, including forecasting tasks, Mean Absolute Error (MAE), Mean Square Error (MSE), or Mean squared logarithmic error (MSLE) are common choices for expressing the loss.

In this paper, we question the assumption that a fixed and generic loss function is in fact the best choice in regression –and more specifically forecasting– problems, and investigate how losses *learned* from experience benefit the model performance in this type of task. Therefore, our work contributes to recent efforts in meta-learning, and specifically to those aimed at so-called *learning to teach* (Wu et al., 2018). In this context, several proposals have been set forth for automatically learning the parameters, components, or shape of loss functions for neural network training. However, as extensively discussed in Section 2, prior studies have focused on parametrizable losses for classification, and little attention has been paid to the learning to teach paradigm in the context of regression tasks. Part of the reason comes from the fact that loss meta-learning has been generally considered an inefficient approach for single-task regression (Sung et al., 2017), under the assumption that losses such as MAE or MSE can already optimally drive regressor training in that case.

Our study challenges such an assumption and shows that, in practical cases, a simple yet aptly designed meta-learning model can in fact *improve* the training of regression models with respect to legacy and presumably optimal losses. As detailed in Section 3, our proposed model, named `MetaLoss`, builds on a joint co-training of the main regressor network and of the loss-learning network, and takes advantage of controlled noise during training to implement the exploration of the correct loss function behavior for rarely observed samples. We apply `MetaLoss` to the specific problem of time series forecasting in Sections 4 and 5, and demonstrate how meta-learning of loss functions can help forecasting tasks under both apparently naive and complex performance metrics. In the case of naive metrics such as MAE or MSE, we disclose how the automated tailoring of the loss to different *magnitudes* (and not only *errors*) of the prediction can provide substantial gains on the accuracy. In the case of complex metrics, we prove that `MetaLoss` can successfully learn differentiable approximations of time-correlated and non-differentiable performance measures observed in two real-world applications, driving the regressor network towards forecasts that improve solutions currently considered in practice.

Overall, by unveiling for the first time the advantages of meta-learning of loss functions for forecasting tasks, our study paves the road for the adoption of this paradigm in a previously unexplored machine learning domain.

## 2 RELATED WORK

Meta-learning, also referred to as learning-to-learn, overcomes the limitations of fixed learning-based models, and allows automatically tuning different aspects of the learning algorithm to the target task (Hospedales et al., 2021). Meta-learning has been successfully applied to, *e.g.*, distillation (Wang et al., 2020), augmentation (Cubuk et al., 2019) or batching (Fan et al., 2018) of training data, initialization (Finn et al., 2017) or optimization (Andrychowicz et al., 2016) of the model parameters, tuning (Micaelli & Storkey, 2020) of its hyper-parameters, and discovery (Liu et al., 2019) of the actual architecture, possibly as a composition of modules (Alet et al., 2019). Our focus is on meta-learning of loss functions, which aims at learning the loss to be used to train the actual model. The problem can be seen as an instance of a hierarchical optimization, where a meta-model is optimized under a constraint represented by the main model optimization (Franceschi et al., 2018). We stress that this is semantically different from meta-learning optimization schedules in iterative and alternate optimization processes (Xu et al., 2019). Specifically, we distinguish two main approaches to loss meta-learning, discussed next.

The first approach consists in having a teacher network infer the most suitable configuration of a predefined, parametrizable loss function. In this direction, several studies have investigated the use of decision networks to select among a set of predefined (family of) loss functions (Liu & Lai, 2020; Denevi et al., 2018), while others have focused on multi-part loss functions, where the goal is setting (Huang et al., 2019; Zhao et al., 2019) and possibly dynamically updating (Heydari et al., 2019) the function weights based on (live) performance metrics. In a similar way, models have been proposed to compose a loss function from primitive mathematical operators (Li et al., 2021), or to express the performance metric as a function of a (reduced) set of simple surrogates (Grabocka et al., 2019; Jiang et al., 2020). Also related to the same concept are strategies such as training a network to correct the optimization trajectory produced by a fixed loss (Huang et al., 2021), or introducing general loss functions that contain hyper-parameters to be learned during training along with the neural network parameters (Barron, 2019). In all these cases, the loss — independently of whether it is expressed as a tunable function, set of primitives, or surrogates — must be designed or selected manually, which is very challenging or even not possible when the performance metric of interest is especially complex, non-differentiable or even not known a priori. With respect to these works, we seek instead a solution that can learn a *clean-slate* loss.

The second approach to loss meta-learning is more appropriate for clean-slate losses, and thus the one we also adopt in our work. The basic concept is representing the loss function itself via a (typically lightweight) neural network, which is fed with relevant input (*e.g.*, main model output, labels, or features) and produces a fit loss to be used to train the main model. In an early work, it was proposed the use of a teacher network to dynamically train parameters of a loss function that adapts to the learning stage of the main model (Wu et al., 2018); yet, this approach still relies on a generic known loss function to be parametrized by the teacher, hence suffers from the same limitations of the studies listed above. Closer to our methodology, the seminal idea of a trainable task-parametrized loss generator was introduced for reinforcement and supervised learning by the meta-critic model, where an action-value function neural network learns to criticise the actions in a specified task (Sung et al., 2017; Zhou et al., 2020). However, the meta-critic model is only applied to supervised learning problems as a tool for pre-training that allows for the few-shot learning of new tasks (*e.g.*, by generalizing to unseen value ranges in the same domain). Indeed, the authors make it explicit that, in the case of a single task, modeling the loss via a dedicated neural network creates an indirection that makes learning more inefficient (Sung et al., 2017). We show instead that loss meta-learning can in fact improve over presumably optimal losses also in simple single forecasting tasks. As such, our model is in fact complementary to the meta-critic one, as `MetaLoss` could be integrated with the meta-critic helper block to automatically model more tasks.

It is worth mentioning that in the vast majority of previous works, and in other studies (Shu et al., 2020; Jiang et al., 2020; Gao et al., 2021; Jiang et al., 2020; Marchetti et al., 2021) that also include recent proposals to employ genetic programming tools to learn loss functions (Gonzalez & Miikkulainen, 2020; 2021), the aim of the loss meta-learning strategy is on discrete-space classification tasks or ranking problems. Little attention has been instead paid until now to loss meta-learning for regression. By investigating meta-learning solutions that target loss functions for forecasting tasks, our work shed new light on the advantages that this emerging paradigm can bring to a class of machine learning problems where loss meta-learning has been overlooked to date.

## 3 A MODEL FOR REGRESSION LOSS META-LEARNING

The proposed approach, named `MetaLoss`, consists of a loss-function-agnostic regressor that performs twofold learning. First, it must learn to forecast the value of action that minimizes a certain loss function. Second, it must also learn which is the said loss function according to the measures obtained from the environment. This concept has obvious applications in many practical regression problems where we can measure the performance resulting from the actions taken by the system, but the expression of the objective cannot be characterized manually because of its complexity, or it is directly not known a priori; in these scenarios, we cannot apply a particular or well-defined loss function during the training phase. However, we will show that the approach can also benefit simpler cases, where the goal is a pure prediction of a time series, and where MAE, MSE, or variants of the same are generally considered to be very effective losses.

In the following, we first provide a formal definition of the problem, and then present the high-level concept of `MetaLoss` as well as the detailed design of its architecture and operation.

### 3.1 PROBLEM FORMULATION

Let us denote the space of system state variables as $\mathbb{S}$. We also denote the input space of the predictor as $\mathbb{X}$ and the output space as $\mathbb{Y}$, such that the predictor can be modeled as $f_{\mathbf{W}^p} : \mathbb{X} \to \mathbb{Y}$, and the decision for time $t+1$ taken at time $t$ by the predictor can be written as[1] $\hat{y}_{t+1} = f_{\mathbf{W}^p}(\boldsymbol{x}_t)$, where $\boldsymbol{x}_t \in \mathbb{X}$ includes the past observations of the system state, and $\mathbf{W}^p$ represents the parameters of the predictor.

Let us denote the performance cost of the predictor's decision taken at time $t$, measured at time $t+1$, as $\mathcal{M}_{t+1} = f_{\mathcal{M}}(y_{t+1}, \hat{y}_{t+1}, \boldsymbol{v}_{t+1})$, where $\boldsymbol{v}_{t+1}$ denotes the current observations (at the time of the measurement $t+1$) of the system variables that may impact $\mathcal{M}_{t+1}$, $y_{t+1}$ denotes the actual value that $\hat{y}_{t+1}$ attempts to predict, and $f_{\mathcal{M}}(\cdot)$ represents the *a priori unknown* expression of the objective. Note that, even if this expression is not known, the performance is assumed to be measurable, and samples of $f_{\mathcal{M}}(\cdot)$ can be obtained by observing the outcome of the predictor's output on the system.

Since the relation $f_{\mathcal{M}}(\cdot)$ is uncharted at first, the model also has to learn it. For that, we define a second optimization function to describe the loss-learning task. Such loss-learning task takes as inputs $(i)$ the predictor's decision $\hat{y}_{t+1}$ and $(ii)$ the current observations $\boldsymbol{v}_{t+1}$, and it casts a performance cost estimate $\tilde{\mathcal{M}}_{t+1} = f_{\mathbf{W}^\ell}(y_{t+1}, \hat{y}_{t+1}, \boldsymbol{v}_{t+1})$, where $\mathbf{W}^\ell$ represents the parameters of the loss-learning process.

Consequently, `MetaLoss` is composed of two optimization problems. First, the loss-learning task aims at correctly characterizing $f_{\mathcal{M}}(y_{t+1}, \hat{y}_{t+1}, \boldsymbol{v}_{t+1})$ through the estimated $f_{\mathbf{W}^\ell}(y_{t+1}, \hat{y}_{t+1}, \boldsymbol{v}_{t+1})$. This is done by computing a legacy loss function (*e.g.*, MAE or MSE). For example, if we consider the MSE $L^2(\boldsymbol{a}, \boldsymbol{b}) = ||\boldsymbol{a} - \boldsymbol{b}||^2$, it follows that the loss-learning optimizer objective is

$$\min_{\mathbf{W}^\ell} \quad L^2\big(f_{\mathbf{W}^\ell}(y_{t+1}, \hat{y}_{t+1}, \boldsymbol{v}_{t+1}), \ f_{\mathcal{M}}(y_{t+1}, \hat{y}_{t+1}, \boldsymbol{v}_{t+1})\big). \tag{1}$$

In turn, the predictor's objective is to minimize the performance cost of its decisions based on the predicted performance $f_{\mathbf{W}^\ell}(y_{t+1}, \hat{y}_{t+1}, \boldsymbol{v}_{t+1})$, *i.e.*, to solve the following optimization:

$$\min_{\mathbf{W}^p} \quad f_{\mathbf{W}^\ell}\big(y_{t+1}, \ f_{\mathbf{W}^p}(\boldsymbol{x}_t), \ \boldsymbol{v}_{t+1}\big). \tag{2}$$

### 3.2 MODEL CONCEPT AND TRAINING

The `MetaLoss` model aims at solving the problem above in an effective manner. To this end, it represents both the predictor and the loss function through Deep Neural Networks (DNN), one for each of the two blocks, as illustrated in Figure 1. Note that `MetaLoss` is a conceptual model that can accommodate diverse implementations: hence, the exact architecture of the neural networks, including their layering and activations, can differ depending on the considered task and the complexity of the

---

[1]Hereinafter, for the sake of simplicity and clarity, we use the scalar notation to describe the predictor's output. Nevertheless, the proposed approach is not limited to uni-dimensional fields and can be applied to more generic multidimensional cases.

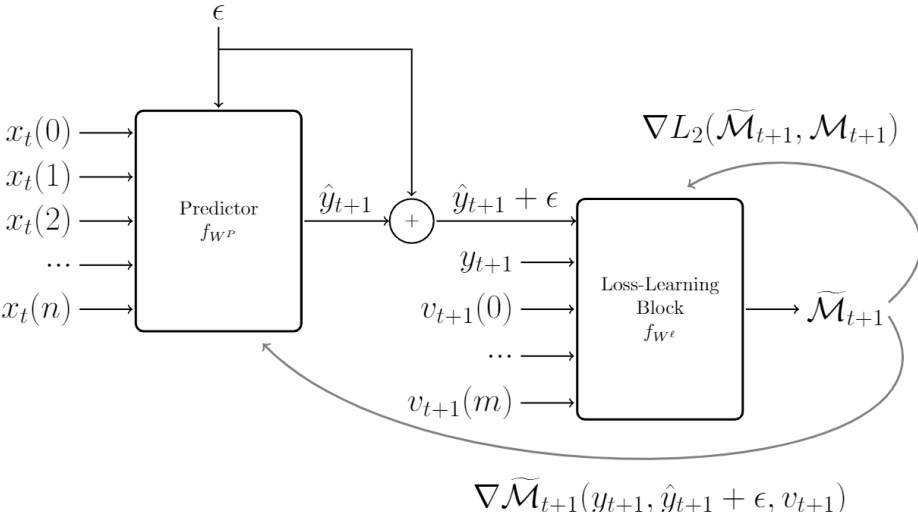

Figure 1: The unknown relation between the objective and the outputs of the predictor is learnt and encoded into a loss-learning block. Then, this block acts as the loss function to train the predictor, such that the predictor directly provides an output fitting the desired metric. The whole process is automated.

involved data. Therefore, the detailed implementation of the two main blocks will be reported in Sections 4 and 5, separately for each application use case.

The algorithm adopted to train the `MetaLoss` model is outlined in Algorithm 1. There, we make use of the Cyclic Learning Rate (CLR) method (Smith, 2017), which lets the learning rate oscillate within a range, with the extreme values of this range updated each batch iteration. This method reduces the sensitivity of the system to the initial configuration, and it prevents that a wrongly selected initial value triggers a poor performance by becoming entrenched in some local minimum. In general, CLR accelerates the automatic convergence, which proved especially useful in contexts where the loss function must be inferred during training. We employ the simple triangular version of CLR, with 5 cycles across the full training phase.

---

**Algorithm 1:** Training procedure of `MetaLoss`

---

Initialize predictor nn, $f_{\mathbf{W}^p}(\boldsymbol{x}_t, \epsilon)$
Initialize loss nn $f_{\mathbf{W}^\ell}(y_{t+1}, \hat{y}_{t+1}, \boldsymbol{v}_{t+1})$
Initialize predictor's learning rate $\alpha_t^p$
**for** $t = \{1, 2, ..., T_{training}\}$ **do**
    Randomly choose $\epsilon$
    Predict the output $\hat{y}_{t+1} = f_{\mathbf{W}^p}(\boldsymbol{x}_t, \epsilon)$
    $\mathbf{W}^\ell \leftarrow \mathbf{W}^\ell - \alpha^\ell \nabla_{\mathbf{W}^\ell} L^2 \big( f_{\mathbf{W}^\ell}(y_{t+1}, \hat{y}_{t+1} + \epsilon, \boldsymbol{v}_{t+1}), f_{\mathcal{M}}(y_{t+1}, \hat{y}_{t+1} + \epsilon, \boldsymbol{v}_{t+1}) \big)$
    $\mathbf{W}^p \leftarrow \mathbf{W}^p - \alpha_t^p \nabla_{\mathbf{W}^p} f_{\mathbf{W}^\ell}(y_{t+1}, \hat{y}_{t+1} + \epsilon, \boldsymbol{v}_{t+1})$
    Update $\alpha_t^p$
**end**

---

Three important remarks are in order, concerning the `MetaLoss` concept and training.

First, the meta-learning model outlined above is able to approximate a non-differentiable objective $\mathcal{M}$ by a *differentiable alternative* $f_{\mathbf{W}^\ell}(\cdot)$, which is implemented by the loss-learning DNN upon training. In turn, this allows optimizing the predictor DNN under metrics that could not be directly used as losses by producing a suitable approximation of the same.

The second remark concerns the fact that the input of the loss-learning DNN is implemented by a different expression than that indicated in the formal problem definition of Section 3.1. Specifically,

as portrayed in Figure 1, instead of providing the computed action $\hat{y}_{t+1}$ to the loss-learning DNN, we feed it with a disturbed version of $\hat{y}_{t+1}$, by adding a random noise $\epsilon$ that is also input to the predictor DNN. As discussed in detail in Section 3.3, this is a methodological novelty that allows *loss exploration* during training.

The third observation is that the `MetaLoss` model design creates the possibility of *co-training* the predictor and loss-learning blocks during a same gradient descent iteration, which allows each DNN to be informed of (and learn from) the improvements of the other: this makes the learned loss $f_{\mathbf{W}^\ell}(\cdot)$ adapted to the inherent forecasting limits of the predictor, and we will expound this aspect in Section 3.4 below.

### 3.3  Loss exploration

During training, the loss-learning block does not receive the exact value output by the predictor, $\hat{y}_{t+1}$, rather it is input a disturbed version of it, $\hat{y}_{t+1} + \epsilon$, where $\epsilon$ is a random variable with zero mean. This is reflected in Figure 1. Therefore, the weights update computed by the gradient descent are shown in Algorithm 1, and are given by the following expressions for a particular weight of the predictor's DNN ($\omega_{t+1}^p$) and a particular weight of the loss-learning DNN ($\omega_{t+1}^\ell$):

$$\omega_{t+1}^p = \omega_t^p - \alpha_t^p \frac{\partial f_{\mathbf{W}^\ell}(y_{t+1},\ \hat{y}_{t+1} + \epsilon,\ \boldsymbol{v}_{t+1})}{\partial \omega_t^p} \tag{3}$$

$$\omega_{t+1}^\ell = \omega_t^\ell - \alpha^\ell \frac{\partial L^2\big(f_{\mathbf{W}^\ell}(y_{t+1},\ \hat{y}_{t+1} + \epsilon,\ \boldsymbol{v}_{t+1}),\ f_{\mathcal{M}}(y_{t+1},\ \hat{y}_t + \epsilon,\ \boldsymbol{v}_{t+1})\big)}{\partial \omega_t^\ell} \tag{4}$$

where $\alpha_t^p$ and $\alpha^\ell$ are the learning rates of the predictor and the loss-learning DNNs, respectively. The dependency of $\alpha_t^p$ on $t$ is due to the use of CLR mentioned in Section 3.2.

The goal of the random variable $\epsilon$ is to allow for further exploration of the input values, supplying the loss-learning block with a broader observation of the input domain beyond that provided by the training samples. This enlargement of the input space improves the reliability of the characterization of the loss function over the continuous domain by the loss-learning block.

Note that the noise $\epsilon$ is only needed during training, and it is set to 0 once the expression of the loss $f_{\mathbf{W}^\ell}(\cdot)$ is learnt, *i.e.*, during model testing. In this regard, a critical design feature of `MetaLoss` is that $\epsilon$ is also input to the regressor DNN: during training, this lets the prediction block learn the correlation between such input and the added disturbance to its output. Then, during inference, setting $\epsilon$ to 0 allows producing forecasts $\hat{y}_{t+1}$ that are not biased by the loss exploration used in training.

### 3.4  Co-training of predictor and loss

`MetaLoss` is implemented as two cascaded DNNs, as illustrated in Figure 1, where the loss-learning block is fed by the current observations and the forecast output by the predictor. This allows us to jointly optimize the two blocks through the same backpropagation process. Specifically, and as also summarized in Algorithm 1, the weights of the two DNNs are optimized during training as follows.

First, during the forward pass, the predictor is fed with a set of past observations of the system state from $N$ previous time instants (as well as other possibly relevant inputs), and it outputs a prediction for $t + 1$ (or, more generally, for the next $M$ next time instants) at time $t$, $f_{\mathbf{W}^p}(\boldsymbol{x}_t, \epsilon)$. At time $t + 1$, the current observations are measured and passed to the loss-learning system, which computes the estimated performance function $\tilde{\mathcal{M}}_{t+1} = f_{\mathbf{W}^\ell}\big(y_{t+1}, f_{\mathbf{W}^p}(\boldsymbol{x}_t, \epsilon) + \epsilon, \boldsymbol{v}_{t+1}\big)$. At the same time, the actual performance of the taken decision $\mathcal{M}_{t+1} = f_{\mathcal{M}}\big(y_{t+1}, f_{\mathbf{W}^p}(\boldsymbol{x}_t, \epsilon) + \epsilon, \boldsymbol{v}_{t+1}\big)$ is measured.

Then, the mismatch between estimated and true performance is evaluated via a legacy or standard loss function, and backpropagated first to the loss-learning DNN. Here, the loss-learning DNN updates its weights $\omega_{t+1}^\ell$ to better capture the relation between $\mathcal{M}_{t+1}$ and the combined values of the prediction $\hat{y}_{t+1}$ and the system state $\boldsymbol{v}_{t+1}$ and $y_{t+1}$. Within the same iteration, the updated loss is sequentially backpropagated to predictor DNN, which allows improving the alignment of the forecast with the optimal decision that minimizes $\mathcal{M}_{t+1}$.

This design increases the efficiency of the training phase with respect to the case where each block is optimized independently, *e.g.*, by feeding the loss-learning block with random predictions and, once the loss has been learned, using it to train the predictor. Indeed, co-training allows learning a loss $f_{\mathbf{W}^\ell}(\cdot)$ that is adapted to the intrinsically limited accuracy of the predictor; as an example, co-training may lead to learning diverse shapes of the loss depending on the magnitude of the target variable $y_{t+1}$ if the quality of the prediction is found to be affected by the absolute value of $y_{t+1}$. We will observe practical situations where this type of adaptation occurs in Sections 4 and 5.

It is worth noting that such a co-training represents a major novelty of our model with respect to previous related proposals (Sung et al., 2017; Wu et al., 2018). Indeed, the end-to-end backpropagation training was not possible in prior models, and the two elements (*e.g.*, the learning-to-act block and the learning-to-correct block) were trained either iteratively or in a nested manner only.

## 4 EXPERIMENTS: TIME SERIES FORECASTING

We first investigate the performance of `MetaLoss` in plain time series forecasting tasks. In these tests, we compare the one-step forecast returned by a same predictor DNN, which is trained under the `MetaLoss` model (*i.e.*, via co-training with a loss-learning DNN), and with a MAE loss. The predictor DNN is identical in the two cases, as all of the hyperparameters are kept the same between the classical MAE and the `MetaLoss` approach; also, the architecture (number and type of layers, number of neurons per layer) are the same, and the same holds for the predictor learning rate, the number of epochs, and the activation functions. The starting seed is also fixed at the start of both trainings to be the exact same in order to reduce randomness as most as possible. The results are presented on MinMaxscaled datasets and averaged with 5 different runs for each experiment.

We remark that the design of the predictor can vary across the forecasting tasks on different datasets later listed in Section 4.1, and we do not detail those here for the sake of brevity. What matters is that the architecture of the predictor DNN is identical under `MetaLoss` and MAE, which allows juxtaposing the results obtained under the two approaches. Concerning the loss-learning block employed by `MetaLoss`, we use for all considered datasets a simple Multi-Layer Perceptron (MLP). The size of the MLP depends on the number of input, *i.e.*, on the size of $v$.

Importantly, we assess the quality of the prediction in terms of MAE itself. Therefore, a static MAE loss that we use as a benchmark for `MetaLoss` represents the apparently optimal –and very commonly adopted– choice to drive the optimization of the predictor.

### 4.1 DATASETS

Experiments are run using different real-world datasets. The first dataset describes the mobile data traffic demand generated by four video streaming services (Facebook Live, Netflix, Twitch, and Youtube) in a large metropolitan area during several consecutive months (Marquez et al., 2017). The data was collected and aggregated by the mobile network operator using passive measurement probes, resulting in traffic levels (in bytes) every five minutes, for a total of more than 22,000 samples for each of the four services.

The second dataset describes the hourly energy consumption in a part of the Eastern Interconnection grid in the United States of America between 2002 and 2018. The data comes from a regional transmission organization (RTO) (Mulla, 2018), and it incorporates information from different power providers, each one them managing a different geographical area. Overall, the time series consists of more than 145,000 samples.

A third dataset describes the energy consumption evolution within a single household over time, using outdoor and indoor characteristics such as temperature, humidity or wind speed as inputs. These data are provided every 10 minutes for more than 4 months, and include over 20,000 data points (Candanedo et al., 2017).

### 4.2 RESULTS

The results for the application of `MetaLoss` to standard loss functions are shown in Table 1 and Figure 2. For each one of the analyzed time series, Table 1 shows the MAE performance (mean and

Table 1: MAE measured in the one-step forecasting of different time series, when a same predictor DNN is trained with a static MAE loss function and with a loss learned via `MetaLoss`. The rightmost column reports the percent gain in MAE scored by the `MetaLoss` approach over the static MAE loss.

| Dataset | MAE $(\times 10^{-2})$ | MetaLoss $(\times 10^{-2})$ | Gain (%) |
|---|---|---|---|
| Facebook Traffic | 2.44±0.05 | **2.33±0.03** | 4.51 |
| Netflix Traffic | 4.38±0.08 | **4.35±0.05** | 0.68 |
| Twitch Traffic | 3.81±0.07 | **3.75±0.04** | 1.57 |
| Youtube Traffic | 3.71±0.03 | **3.68±0.03** | 0.81 |
| Power Grid | 2.68±0.04 | **2.55±0.03** | 4.85 |
| House Energy | 2.33±0.03 | **2.27±0.02** | 2.58 |

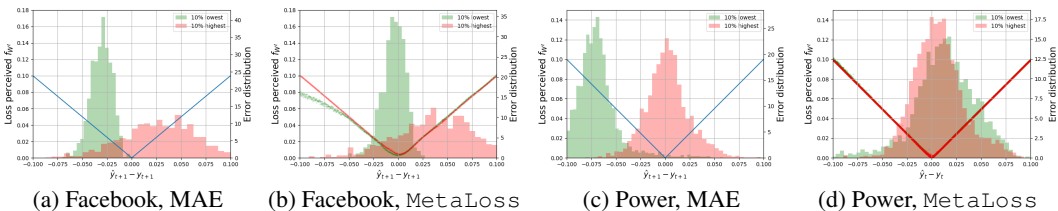

(a) Facebook, MAE     (b) Facebook, `MetaLoss`     (c) Power, MAE     (d) Power, `MetaLoss`

Figure 2: Loss function and error distributions for the top and bottom 10% of predicted values, for the Facebook Live traffic under (a) MAE loss and (b) `MetaLoss`, and for the energy demand under ca) MAE loos and (d) `MetaLoss`.

standard deviation) obtained from $(i)$ training the predictor block with standard MAE loss function (leftmost column), and $(ii)$ using the approach `MetaLoss` presented above, where the loss function is also learned (central column), as well as the percentage gain of `MetaLoss` over standard learning. Table 1 shows how `MetaLoss` reduces costs for standard loss function as MAE up to 5% in the best cases observed. This represents a significant gain, considering that the predictor architecture is the same for a MAE loss and for our model, especially if we take into account that `MetaLoss` is trained to learn the MAE loss function. Thus, `MetaLoss` succeeds in *learning more than the teacher knows*.

This excellent performance mostly comes from the fact that `MetaLoss` adapts its shape to the input data and, therefore, it optimizes the training phase almost independently for different input values. We can clearly see this phenomenon happening in plots (c) and (d) of Figure 2, where the loss learned by `MetaLoss` has learned and evolved during training in order to minimize the loss for both sets represented (the 10% highest and 10% lowest value samples), whereas with the standard model one of these sets is clearly biased at the end of the training phase. This can also be seen from a different perspective within the case of Facebook Traffic (plots (a) and (b)): Although the bias correction is only visible to a lesser extent, we can clearly see how the loss learning function has actually a different shape for each one of the two sets considered.

## 5    EXPERIMENTS: APPLICATION USE CASES

Having proven the advantage that `MetaLoss` can yield in the context of plain time series forecasting tasks that aim at minimizing the MAE of the one-step prediction, we consider the more convoluted case where the relation between the decision space output by the predictor and the objective performance is a complex, non-differentiable, and possibly not (fully) known function of the prediction. We analyze two different use cases with strong practical applications, *i.e.*, $(i)$ anticipating the resources to be allocated in a mobile network to serve real-world traffic demands, and $(ii)$ managing power grid settings to serve the energy demand in a nationwide scenario. These use cases employ the first two datasets outlined in Section 4.1, and are characterized by entangled and diverse relations between the prediction and the performance metric, as detailed next.

Table 2: Operator cost of an anticipatory allocation of network resources measured for the mobile data traffic generated by four different services, when a same predictor DNN is trained with a static $\alpha$-OMC loss function (Bega et al., 2019) and with a loss learned via `MetaLoss`. The rightmost column reports the cost reduction achieved by the `MetaLoss` approach over the static $\alpha$-OMC loss.

| **Dataset** | $\boldsymbol{\alpha}$**-OMC** $(\times 10^{-1})$ | `MetaLoss` $(\times 10^{-1})$ | **Gain (%)** |
|---|---|---|---|
| Facebook Traffic | 1.37±0.06 | **1.26±0.04** | 8.03 |
| Netflix Traffic | 1.57±0.09 | **1.52±0.07** | 3.29 |
| Twitch Traffic | 1.53±0.07 | **1.47±0.05** | 3.92 |
| Youtube Traffic | 1.36±0.05 | **1.31±0.04** | 3.68 |

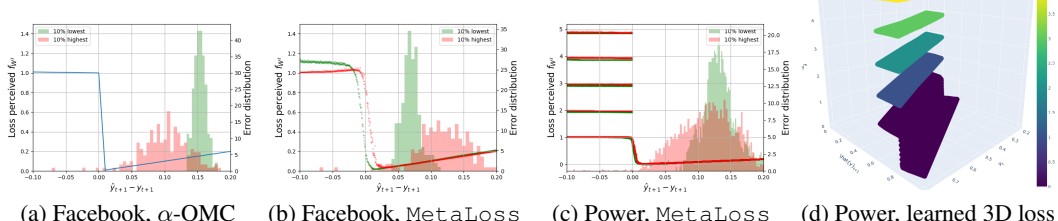

(a) Facebook, $\alpha$-OMC  (b) Facebook, `MetaLoss`  (c) Power, `MetaLoss`  (d) Power, learned 3D loss

Figure 3: Loss function and error distributions for the top and bottom 10% of predicted values, for the network resource allocation to the Facebook Live service under (a) $\alpha$-OMC loss and (b) `MetaLoss`, and for the power grid management with `MetaLoss`, with a (c) two-dimensional and (d) three-dimensional representation of the learned loss.

## 5.1 USE CASE I: ANTICIPATORY NETWORK RESOURCE ALLOCATION

**Metric.** The goal of the operator at time $t$ is forecasting the required network resources (*e.g.*, data transport capacity) needed to serve the demand for each of four streaming services (Facebook Live, Netflix, Twitch, and YouTube) in the following 5-minute time step. A plain prediction is insufficient in this case, as the cost is asymmetric: specifically, the operator seeks to $(i)$ avoid an expensive monetary fee $\beta$ owed to the service provider in case insufficient resources are allocated and the future traffic $y_{t+1}$ cannot be served, and $(ii)$ prevent unnecessary overdimensioning beyond $y_{t+1}$ in case of prediction errors with a positive sign. Formally, the cost incurred by the operator is

$$\mathcal{M}_{t+1} = \beta \cdot \mathbf{1}_{\hat{y}_{t+1} < y_{t+1}} + (\hat{y}_{t+1} - y_{t+1}) \cdot \mathbf{1}_{\hat{y}_{t+1} \geq y_{t+1}}. \tag{5}$$

This expression is clearly not differentiable due to the presence of the indicator functions $\mathbf{1}_C$, which takes value 1 if the condition $C$ in the subscript is met, and takes the value 0 otherwise. Recent studies in the computer networking literature have proposed an expert-designed loss function, termed $\alpha$-OMC, which is a manually devised differentiable approximation of Equation (5) above (Bega et al., 2019). We employ $\alpha$-OMC as a state-of-the-art fixed loss benchmark to be used as a term of comparison for `MetaLoss`.

**Network architecture.** In this experiment, we employ as the predictor $f_{\mathbf{W}^p}$ a multi-layer RNN regressor. The loss-learning neural network $f_{\mathbf{W}^\ell}$ is an MLP with the simplest possible case for `MetaLoss`, using only $y_{t+1}$ and $\hat{y}_{t+1}$ as input. The predictor RNN is also trained using the $\alpha$-OMC loss function for comparison purposes. Results are provided in the Table 2 for the four services.

**Results.** We can see in Table 2 how `MetaLoss` yields cost reductions over $\alpha$-OMC that range from 3% to 8%, which correspond to significant operating expense cuts in mobile network infrastructure management. This is made possible by the fact that `MetaLoss` learns a loss that is better tailored to the performance metric in Equation 5 than a human-designed version developed with full knowledge of the metric itself. In a sense, the result proves how meta-learning losses for regression via `MetaLoss` allows *apprehending more than the teacher knows*. Indeed, co-training the predictor and loss-learning block as done in our proposed model enables discovering a loss that is optimized for different absolute values of the predicted variable.

This adaptation is illustrated in the first two plots in Figure 3, where the losses and error distributions are told apart for the $10\%$ of cases with lowest and highest traffic demands, under (a) the $\alpha$-OMC loss and (b) `MetaLoss`. While the learned loss in `MetaLoss` always captures the general behavior of Equation 5, we can observe slight shifts in the error $\hat{y}_{t+1} - y_{t+1}$ that minimizes the cost, depending on the absolute value of $y_{t+1}$. In other words, `MetaLoss` learns a loss that naturally compensates for the different accuracy of the predictor in anticipating traffic values of diverse magnitude. Such an adaptation is impossible to ascertain by just looking at the performance metric (*i.e.*, the "teacher"), as it inherently depends on the prediction quality; yet, it is successfully learned by our model.

## 5.2 Use case II: Power grid management

**Metric.** The complex field of power grid management and the study of smart grids are ruled by a significant number of diverse Key Performance Indicators (KPI) (Pow, 2012; Personal et al., 2014; Harder, 2017). One of the fundamental dimensions that define the performance in such scenarios is the reliability of the network, *i.e.*, how often the network fails to provide the required power.

Interestingly, the reliability in power management is not only measured by the *frequency* of power cuts due to the under-provisioning, but also by the *duration* of these cuts (Pow, 2012). The service provider is especially interested in preventing under-estimations, as in the previous use case of network resource allocation; however, the metric applied to capacity forecasting cannot be considered to this scenario because it does not take into account the duration of the under-provisioning. Now, we require a cumulative metric that incorporates some memory. Hence, the metric considered for the Power Grid Management use case is as follows:

$$\mathcal{M}_{t+1} = (\beta + \mathcal{M}_t \cdot \mathbf{1}_{\hat{y}_t < y_t}) \cdot \mathbf{1}_{\hat{y}_{t+1} < y_{t+1}} + (\hat{y}_{t+1} - y_{t+1}) \cdot \mathbf{1}_{\hat{y}_{t+1} \geq y_{t+1}}. \tag{6}$$

We can see that, in the case of overdimensioning, the cost scales linearly with the unnecessary estimated power as for the previous use case. Yet, the cost is considerably different in case of underestimation: if the previous forecast was also underestimated, the current cost is added to the precedent cost in a recursive manner. Thus, this metric depends on the previous state of the network.

**Network architecture.** In this experiment the predictor $f_{\mathbf{W}^p}$ used is also a multi-layer RNN regressor. The loss neural network $f_{\mathbf{W}^\ell}$ is a MLP using 3 inputs, *i.e.*, $y_{t+1}$ and $\hat{y}_{t+1}$, plus the number of previous successive under-provisioned samples $v$ (maxed at 5), which is a time-dependent variable. Designing a loss function matching this metric is nothing simple and would not even make sense. `MetaLoss` allows to use such metric which would not be possible otherwise. As 3 values are used as input of the loss neural network $f_{\mathbf{W}^p}$, the resulting loss function is a 3 dimensional shape in a 4-dimension space. This resulting loss after training is presented in plot (d) of Figure 3, where the color describes the $v$ dimension and where only discretized values are presented corresponding to the number of successive past underprovisioned samples.

**Loss-learning results.** For this use case, we are particularly interested in showing how `MetaLoss` can adapt to complex cost relations. This can be observed in plots (c) and (d) of Figure 3: `MetaLoss` precisely learns the complex and recursive cost function, which depends on previous samples. The result shows the potential of the proposed approach to characterize unknown and non-trivial loss functions for generic regression problems.

## 6 Conclusions

We have proposed `MetaLoss`, a meta-learning model for regression losses, and unveiled how this previously overlooked approach can in fact yield significant gains in forecasting tasks. Experiments with heterogeneous datasets prove how `MetaLoss` can improve the performance of plain one-step predictors in minimizing the MAE of the forecast, with respect to the common practice of employing a fixed MAE loss. Similarly, our tests demonstrate how `MetaLoss` can successfully learn losses that capture complex, time-correlated and non-differentiable metrics. Finally, we highlight for the first time how `MetaLoss` co-training of prediction and loss allows *learning more than the teacher knows*, since the loss can be tailored to the inherent (in)accuracy of the predictor, *e.g.*, for different values of the target variable.

ETHICS STATEMENT

In our work, we employ de-personalized datasets that report information on mobile service usage and energy consumption aggregated over millions of individuals. This makes re-identification of data subjects from the data impossible, hence our research do not involve risks for the mobile subscribers or power grid customers.

REPRODUCIBILITY STATEMENT

We commit to make the source code of `MetaLoss` publicly available upon publication of the paper, so as to ensure the reproducibility of our results. Also, the power grid and house energy datasets are openly accessible at (Mulla, 2018) and (Candanedo et al., 2017), respectively, which will allow regenerating the results in plots (c) and (d) of Figures 2 and 3.

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
