# OpenReview forum: "Loss meta-learning for forecasting"
_ICLR.cc/2022/Conference — ICLR 2022 Submitted_

### Official Review · Reviewer_9TL4 · 2021-10-28

**Correctness:** 2
**Technical Novelty And Significance:** 2
**Empirical Novelty And Significance:** 1
**Recommendation:** 3
**Confidence:** 4

**Main Review:**

Strengths

- The motivation of this work is clear and the authors have covered the relevant literature adequately.

Weaknesses

-  Authors claim that the proposed MetaLoss is a loss-function agnostic framework (Page 3), however in the experiments, they have only used MAE as the loss function in the loss-learning block to evaluate the model. To claim MetaLoss is loss-function agnostic, authors must evaluate this framework using multiple loss functions such as MSE, Huber loss, Tweedie loss (can evaluated using intermittent time series data).

- Moreover, authors have used only one prediction architecture (DNN) to train with the MetaLoss. This raises doubts over the extensibility of the proposed framework, whether MetaLoss can be used to train with other prediction architectures such as RNN, CNN, and tree based regressors such as LightGBM. This because these architectures are currently being heavily used in the time series forecasting literature; therefore, to assess the aptness of this method in the time series forecasting context, it is important to evaluate the proposed Metaloss method with these architectures. If these evaluations can’t be done, authors should clearly explain the limitations of this study (why these experiments can’t be done). Moreover, authors have not mentioned about the hyper-parameter ranges used to train the DNN and nor the procedure used to fine tune the hyper-parameters (at least using an appendix)

- In the experiments, authors mention that they generate one-step forecasts for each time series. Does that mean you generate one step ahead forecasts, i.e., forecast horizon is equivalent to 1, What is the reason for that? Generally, in time series forecasting, we generate multi-step ahead forecasts, i.e., for multiple horizons. Also, the benchmark datasets used in the experiments are also not commonly used datasets in the forecasting literature (check [1] for more commonly used datasets)

-  Another concern is that the study uses MAE as the primary loss function in the experiments and then evaluate the accuracy of the generated forecasts using MAE error measure again. What is the reason for this? This is not the general approach used in time series forecasting, because once the machine learning models are trained using the loss functions such as MAE, MSE, we use proper forecasting error measures such as sMAPE, MASE to evaluate the forecasts. Moreover, authors also do not show whether the result differences are statistically significant (showing the percentage of improvement is not adequate)

-  My biggest concern is about the computational cost of the proposed framework. This study does not report the overall running time of the MetaLoss and compare it against the classical fixed learning-based models. Without the running times, it is quite difficult to conclude the usability of this framework for more data extensive applications.

[1] https://forecastingdata.org/


**Summary Of The Paper:**

The study proposes MetaLoss, a loss meta-learning approach that is capable of autonomously optimising the properties of traditional loss functions (e.g., MAE, MSE) to a target task. The proposed framework has improved the regression accuracy in multiple time series datasets, outperforming classical fixed learning-based models, which do not employ any learning procedure to train a target loss function. The motivation of this work is clear and sound; nevertheless, I have multiple concerns about the experimental setup of this work.

**Summary Of The Review:**

Even though the motivation of this work is clear, I have multiple concerns about the scientific rigour of the experimental setup (see the review section). Based on the current status of the manuscript, I am recommending to reject this paper.

---

### Official Review · Reviewer_5PNy · 2021-10-29

**Correctness:** 3
**Technical Novelty And Significance:** 3
**Empirical Novelty And Significance:** 3
**Recommendation:** 6
**Confidence:** 3

**Details Of Ethics Concerns:**

None.

**Main Review:**

# Strengths

Conversion of the non-differentiable loss function to a differentiable loss function has been a tough challenge. The authors have proposed a meta-learning of loss approach for this task. Their proposal has been clearly explained and well supported through experiments and discussion.

# Weaknesses

(i) The paper could have been written well for readability. I find that the wording was quite confusing at times, thereby breaking the flow.

(ii) Since “loss exploration” and “co-training” is claimed as a novelty I think there need to be some experiments displaying their effect for when they are or aren’t used. It might be that these ideas haven’t been explored in the “meta-learning of loss” community. However, it still warrants more discussion around what are the benefits these proposals give. E.g.

- How does $\epsilon$ help the learning task? Perhaps, further experiments can be conducted to help understand how $\epsilon$ is helping achieve “loss exploration”. What happens when there is no “loss exploration”? How does MAE compare to MetaLoss in the presence or absence of this noise? Does $\epsilon$ depend on your dataset?
- Co-training is more understandable, in my opinion. However, it might be useful to show the effects of CLR or no-CLR on the performance of MetaLoss. For example, how does the metric performance vary with different schedules of CLR (or no-CLR)?

(iii) *MetaLoss succeeds in learning more than the teacher knows* - I find this sentence/claim to be misleading/confusing. Consider, for example,  (A) MAE loss and (B) loss-learned NN. (A) knows nothing about data, while (B) adapts itself to the data and the predictor. Thus, "teacher knows" only applies to (B) while, if I understand correctly, it is meant to refer to (A), thereby causing the whole confusion. (B) uses (A) as a meta-teacher and adapts itself to help the predictor perform well on (A).

(iv) Are there examples of plain regression tasks (not time-series forecasting) on which MetaLoss's performance can be demonstrated? The authors have presented this model as generically applicable to regression, however, the experiments are conducted on time-series forecasting tasks. It might be useful for the readers to understand the effect of MetaLoss on vanilla regression tasks.

*Clarifications* -
- How are the points for Loss Perceived in figures obtained for MetaLoss? How are y_t and \hat{y}_{t} decided for the loss-learning DNN?


*Minor Comments*
- Figure 2: typo - “loos”
- “where the loss learned by MetaLoss has learned and evolved during training “ - confusing phrasing of the sentence
- Figure 2 is quite confusing. It talks about evolution, but there is no notion of time. Also, the y-axis is not well explained, e.g., “Perceived loss”.
- If “Overdimensioning” is the same as overestimation, it might be helpful to use this term as it's more familiar to the readers.
- Figure 3(d) labels are not clear



**Summary Of The Paper:**

This work proposes meta-learning of loss functions for regression tasks. This approach has powerful applications for cases when the performance metrics are non-differentiable. Such cases often arise in tasks where the cost of misprediction is asymmetric and ripe with conditional clauses. The authors have proposed a simple yet powerful two-step process, wherein two neural networks are involved such that (a) a loss-learning NN learns an approximate differentiable loss function that mimics the non-differentiable performance metric, and (b) predictor NN that aims to perform well on the loss function defined by this loss-learning NN. The authors specify two crucial approaches to make this proposal work - (a) loss exploration, and (b) co-training of two neural networks. Their proposal is well supported by the experiments on time-series prediction tasks.

**Summary Of The Review:**

The model proposed by the authors seems to have powerful use cases. Their proposal has been well explained and supported by experiments. I would love to see some more experiments/discussions around some of their claims. I am happy to reconsider my scores thereafter.

---

### Official Review · Reviewer_qQ83 · 2021-11-01

**Correctness:** 1
**Technical Novelty And Significance:** 2
**Empirical Novelty And Significance:** 1
**Recommendation:** 3
**Confidence:** 5

**Main Review:**

The meta-learning problem studied in this work is to learn the loss function used in a time-series forecasting method. An intuitive overview figure that succinctly illustrates the problem would be extremely useful for the reader in Section 3.1 of the problem formulation. The symbol and terms denoted in the first sentence of the problem formulation are never used in the actual problem, “Let us denote the space of system state variables as S.”

The evaluation of the method in Section 4 is not convincing. To demonstrate the effectiveness of the approach, it is required to compare to the state-of-the-art methods across all datasets. However, the results provided in Table 1 of this paper uses only a single very simple baseline for comparison. To be able to understand the advantage of this method, MetaLoss should be compared more fairly with state-of-the-art methods. There are also other issues. For instance, the training runtime is also not shown, which would obviously be much larger for MetaLoss.

Results are also for a single selection of hyperparameters, which includes number and type of layers, number of neurons per layer, predictor learning rate, number of epochs, and activation functions, and so on. It would have been useful to see an ablation study where many of these are varied and the performance is shown. Currently, I didn’t even notice where the actual values used in the one experiment are given, so it seems impossible to even reproduce the simple result shown in Table 1. Nevertheless, these need to be stated, and an ablation study showing performances for a variety of settings need to be provided, otherwise, it is impossible to know whether the method works for only one specific selection of hyperparameters, or if it works better for other possible settings, and when does it work well and when does it fail. Currently, we only see results for one specific setting from this enormous space of hyperparameters (which are also seemingly missing from the paper). This is hardly proof that the method works well, or is even useful. It is probably easy to find a selection of hyperparameters where it gives worse results, given the huge space of possibilities. This appears to be a major flaw in the evaluation.
Runtime performance is also crucial. In many applications, having a faster approach may be more important than performance, and thus may be willing to trade-off a minor decrease in performance for a significantly faster approach.

The writing of the paper also requires significant improvement, as various parts are unclear. The problem is not well motivated, and the approach can also be explained significantly better with more concrete and succinct examples.

Typo in first paragraph of Section 4, “… to reduce randomness as most as possible.”
Also, in the caption of Figure 2, “MAE loos”. Some parts of Figure 2 are also not explained.

The paper does a poor job on reproducibility, as mentioned above. The hyperparameters used in the one experiment are not provided, nor is it explained why these are chosen, etc. Source code is also not available, and the majority of the data (the 4 traffic datasets) appear to also be private (as far as I can tell). Hence, the first sentence of the reproducibility statement itself seems incorrect, “… so as to ensure the reproducibility of our results”

There are some terms used in the paper incorrectly or in a misleading way, e.g., “we prove” on page 1. Many of these sentences also are somewhat overclaiming due to the above issue of showing results for a single hyperparameter selection, which may have been difficult to find and cherry picked. It is unclear if the approach actually works better for other selections, even against this simple baseline.



**Summary Of The Paper:**

This paper focuses on meta-learning of loss functions for time-series forecasting, where the loss function to be used to train the actual model is learned. For this problem, they propose MetaLoss, which performs joint co-training of a regressor network and of the loss-learning networks. They show under one specific set of hyperparameters and for a single simple baseline method, that it can work well. They also apply it for a few different applications. The idea seems interesting. However, the actual contribution of this work, and effectiveness of the approach is unclear due to the issues discussed in detail below.

**Summary Of The Review:**

While the idea of MetaLoss seems interesting, the evaluation and comparison of it is not convincing and needs significant work. For instance, instead of comparing a single simple baseline to MetaLoss, the approach needs to be compared to the state-of-the-art forecasting methods (e.g., NBEATS, Deep Factors, Graph Deep Factors, DeepAR, and so on…). The authors should also include experiments showing the runtime performance across all these baseline methods and datasets. Such results are omitted in this work, though are crucial.

The exact hyperparameters used are also seemingly missing from the paper. But most importantly, the results shown in the paper are for a single set of hyperparameters, and therefore, it is unclear if the model works better for the specific hyperparameters chosen, or if it routinely outperforms the other across a wide variety of hyperparameter selections. In other words, the results shown are not significant due to this issue. It is likely easy to find a selection of hyperparameters where the proposed approach performs poorly compared to even the simple baseline shown.

Furthermore, the writing can also be significantly improved including the introduction, problem formulation, and motivation and importance of the problem. Overall, this paper needs significant work before it can be accepted. See above for other important issues that need to be addressed as well.

---

### Official Review · Reviewer_m1fA · 2021-11-07

**Correctness:** 2
**Technical Novelty And Significance:** 1
**Empirical Novelty And Significance:** 1
**Recommendation:** 1
**Confidence:** 4

**Main Review:**

In my opinion, this paper has the following three major problems.

1. Unlike the vast majority of meta-learning approaches where the outer-learning objective is set to the validation or test loss, In this paper the outer learning objective simply aim to minimize the gap from the existing loss function. This seems very weird to me. The goal of loss learning is to outperform the existing loss functions such as MAE, but they propose to minimize the gap between MAE and their loss learning module. How does it make sense? Did the authors justify this choice in the paper rigorously? I think this is why they introduce injecting noise during the inner- and outer-learning process to make the loss learning non-trivial, but it also seems very ad-hoc and no intuition has been provided as to why doing so. This is very a significant methodological concern which requires thorough explanations.

2. In section 3.4, they argue that their method of co-training predictor and loss is novel, but this is completely wrong. As far as I understood, this "co-training" is almost identical to one-step lookahead approach (i.e. T1-T2) or online first-order approximation widely used in many meta-learning and gradient-based hyperparameter optimization literatures. There exists tones of papers doing so, but they fail to recognize it. The authors need to cite them and clearly specify what is different from those existing literatures.

3. Lastly, there is just too little baselines. At least, the authors should have compared against MSE, MSLE, and other existing pre-defined loss functions. The authors also should justify other regularizers such as dropout or smoothing techniques in order to justify that the performance improvement is not simply due to such randomness. Most importantly, the performance improvement is very marginal. I guess this is why they also report what they call "Gain(%)", which also seems very weird to me. Is it a standard way of reporting performance in other literatures?

**Summary Of The Paper:**

This paper proposes a loss learning framework for the timeseries forecasting regression problem. They introduce two learning blocks, one for producing the next step prediction and the other for learning the loss function. The experimental results show that the proposed method slightly increase the performance compared to MAE.

**Summary Of The Review:**

In my view, this paper is well below the acceptance threshold and more literature survey seems required.

---

### Decision · Program_Chairs · 2022-01-20

**Decision:**

Reject

**Comment:**

The meta-learning framework based on learning the loss function for time series forecasting is an interesting and important topic. However, the reviewers think the literature, baselines, and experimental results need significant improvement.